Corrected: Author correction

# Engineering Auger recombination in colloidal quantum dots via dielectric screening

Xiaoqi Hou[1], Jun Kang [2], Haiyan Qin [1], Xuewen Chen[3], Junliang Ma[1], Jianhai Zhou[1], Liping Chen[1], Linjun Wang [1], Lin-Wang Wang[2] & Xiaogang Peng [1]

Auger recombination is the main non-radiative decay pathway for multi-carrier states of colloidal quantum dots, which affects performance of most of their optical and optoelectronic applications. Outstanding single-exciton properties of CdSe/CdS core/shell quantum dots enable us to simultaneously study the two basic types of Auger recombination channels— negative trion and positive trion channels. Though Auger rates of positive trion are regarded to be much faster than that of negative trion for II-VI quantum dots in literature, our experiments find the two rates can be inverted for certain core/shell geometries. This is confirmed by theoretical calculations as a result of geometry-dependent dielectric screening. By varying the core/shell geometry, both types of Auger rates can be independently tuned for ~1 order of magnitude. Experimental and theoretical findings shed new light on designing quantum dots with necessary Auger recombination characteristics for high-power light-emitting-diodes, lasers, single-molecular tracking, super-resolution microscope, and advanced quantum light sources.

[1] Center for Chemistry of Novel & High-Performance Materials, and Department of Chemistry, Zhejiang University, 310027 Hangzhou, People's Republic of China. [2] Material Science Division, Lawrence Berkeley National Laboratory, Berkeley, CA 94720, USA. [3] School of Physics, Huazhong University of Science and Technology, 430074 Wuhan, People's Republic of China. Correspondence and requests for materials should be addressed to H.Q. (email: hattieqin@zju.edu.cn) or to L.-W.W. (email: lwwang@lbl.gov) or to X.P. (email: xpeng@zju.edu.cn)

olloidal quantum dots (QDs) are semiconductor nano-
crystals with their sizes in the quantum-confinement
regime[1]. For multi-carrier states—charged excitons and
bi-/multi-excitons—of QDs, Auger recombination consumes the
energy of a photo- or electro-excited electron-hole pair (exciton)
to promote another carrier to a high-energy level[2,3]. Auger
recombination is the main non-radiative energy dissipation
pathway of multi-carrier states, affecting all aspects of their
properties[4]. Because of relaxation of momentum conservation
requirement and increased Coulomb interactions, Auger recom-
bination is very efficient in QDs in comparison with the corre-
sponding bulk semiconductors[3,5]. Different from small-molecular
emitters, multi-carrier states are quite commonly generated in
QDs under either photo-excitation[3,6–8] or electro-excitation[9,10].
For instance, due to Auger non-radiative recombination of multi-
carrier states, short gain lifetime in QD lasing[6,8] and efficiency
roll-off in QD-based light-emitting-diodes[10,11] are frequently
observed. As the wavefunction of a band-edge hole (or electron)
is mostly confined to the inner portion of a core/shell QD,
charged QD is usually quite stable. Charging leads to photo-
luminescence blinking of single QD—random switching of pho-
toluminescence between a dim charged state with efficient Auger
non-radiative recombination and a bright neutral state[12–14].
Photoluminescence blinking brings issues for certain applications,
such as single-molecule tracking in biological systems[15,16]. It is
interesting to note that Auger recombination is not always det-
rimental. For instance, photoluminescence blinking of QDs can
be applied for super-resolution microscopy[17], and Auger is ben-
eficial for purity of QD-based single-photon source powered by
either photo-excitation[18–20] or electro-excitation[21]. These facts
suggest that the challenge is how to tune Auger non-radiative
recombination of QDs in a controllable manner.

Synthetic chemistry of QDs recently makes ideal single-exciton
properties be accessible for the most developed CdSe/CdS core/
shell QDs[14,22,23], but synthetic control of multi-carrier states
remains in its infancy primarily because of poor understanding
of Auger non-radiative recombination in QDs[24]. Basic forms of
multi-carrier states in a QD are negative and positive trions that
are, respectively, composed of one exciton plus one extra electron
or hole[4]. Interestingly, Auger non-radiative recombination of
other multi-carrier states, such as biexciton and multi-exciton,
can be treated as various combinations of negative- and positive-
trion Auger pathways[5,25,26], simplifying all Auger recombination
processes into two basic channels.

Continuous efforts have been devoted to study and tune Auger
recombination in QDs in the past 20 years. Based on existing
experimental results[27–30], Klimov et al. summarized three stra-
tegies for Auger engineering, namely control of the overlap of
electron and hole envelop wavefunctions, control of conduction-
and valence-band mixing, and control of strength of the intra-
band transition[4]. Although tuning Auger recombination of
negative trion of QDs has been studied widely[28,31–34], behavior of
Auger recombination of the positive trion is largely unknown.
Furthermore, for commonly studied CdSe/CdS core/shell[25,26,31],
CdSe/gradient-shell[25,26], and other II–IV QDs[28], Auger recom-
bination of positive trion has been regarded to be always dom-
inating that of the corresponding negative trion.

Using CdSe/CdS core/shell QDs as a model system, we show
independent assessment of both basic Auger recombination
channels by separately tuning dimensions of the CdSe core and
CdS shell. Our results demonstrate that, for certain geometric
structures, Auger recombination of positive trion can actually
become significantly slower than that of the corresponding
negative trion. By altering the core and/or shell dimensions, rates
of both basic Auger recombination channels of CdSe/CdS core/
shell QDs can be readily tuned by ~ 1 order of magnitude. Besides

the known factors applied for Auger engineering for QDs[4], the-
oretical calculation demonstrates that the unexpected phenomena
are associated with geometry-dependent dielectric screening.
Given the critical role of Auger recombination for QDs, the
experimental and theoretical results in this work should open a
new door to design and synthesize QDs with desired optical and
optoelectronic properties.

## Results

**Optical properties of single-exciton and biexciton.** Previous
reports indicate that properties of single-exciton in QDs are cri-
tical for reliably determining Auger recombination rates of multi-
carrier states[35,36]. Here, CdSe/CdS core/shell QDs with nearly
ideal single-exciton properties are synthesized with various core
sizes (3–7 nm) and shell thicknesses (4–10 monolayers) (Sup-
plementary Fig. 1). For simplicity, this report adopts a systematic
abbreviation, in which CdSe$_{###}$/$x$CdS represents the first-exciton
absorption peak of CdSe core at "###" nm with "$x$" monolayers of
epitaxial CdS shell. For their single-exciton states, these QDs all
possess nearly unity photoluminescence quantum yields (QY),
similar photoluminescence peak widths for ensemble and single
dots, and mono-exponential photoluminescence decay dynamics
with similar lifetimes at both ensemble and single-dot levels
(Fig. 1a, b, Supplementary Fig. 2, Supplementary Table 1, Sup-
plementary Table 2, and Supplementary Note 1). With such QDs,
the radiative and Auger non-radiative recombination rates of
multi-carrier states could be readily and reliably determined by
photoluminescence decay dynamics and photoluminescence QYs
of the given states following existing protocols (see Supplemen-
tary Note 2).

With pulsed laser excitation at low power, area ratio between
the center and side peaks ($g_0^{(2)}$) of the second-order photon
intensity correlation curve of single QD equals to the biexciton
QY, given the single-exciton QY being unity[37]. Alternatively,
biexciton QY has also been determined by emission saturation
curve of either single-dot[38,39] or ensemble samples in solid thin
film[40]. The first approach becomes not reliable when $g_0^{(2)}$ is either
too small or larger than 0.5. The second approach requires the
dots simultaneously to be uncharged, anti-bleaching, and with
equal excitation intensity for all dots under high excitation power,
which are challenging for either single QD or solid-film ensemble
measurements. A method using micro-liquid film of QDs is
introduced here for saturation measurements (Fig. 1c), which can
overcome issues of saturation measurements with either single-
dot or ensemble samples in thin solid-film (Supplementary Fig. 3
and Supplementary Note 3 for experimental details). All
saturation curves obtained using this method (see Fig. 1d for
example) begin to deviate from the Auger-free line when <N>,
the average number of photons absorbed per excitation pulse,
increases to ~ 0.5, which corresponds to <10% QDs at bi- or
multi-exciton states assuming a Poisson distribution of N. Along
with the emission-intensity deviation from the Auger-free line,
the corresponding photoluminescence decay dynamics of the
QDs gradually shows a fast-lifetime component, in addition to
the mono-exponential photoluminescence decay of the single-
exciton (Fig. 1 and Supplementary Figs. 4 and 5)[26,32]. In contrast
to linear dependence of the single-exciton emission on excitation
power, the integrated emission intensity of the fast-lifetime
component follows quadratic power law (Fig. 1f). These results
identify the fast-lifetime channel is associated with the biexciton
emission rate ($k_{XX}$).

Figure 1d shows that all saturation curves of QD emission
intensity can be well fitted using the standard saturation model
(Supplementary Fig. 6 and Supplementary Note 4), which yields
biexciton QY ($QY_{XX}$) with high accuracy[38,39]. For the QDs with

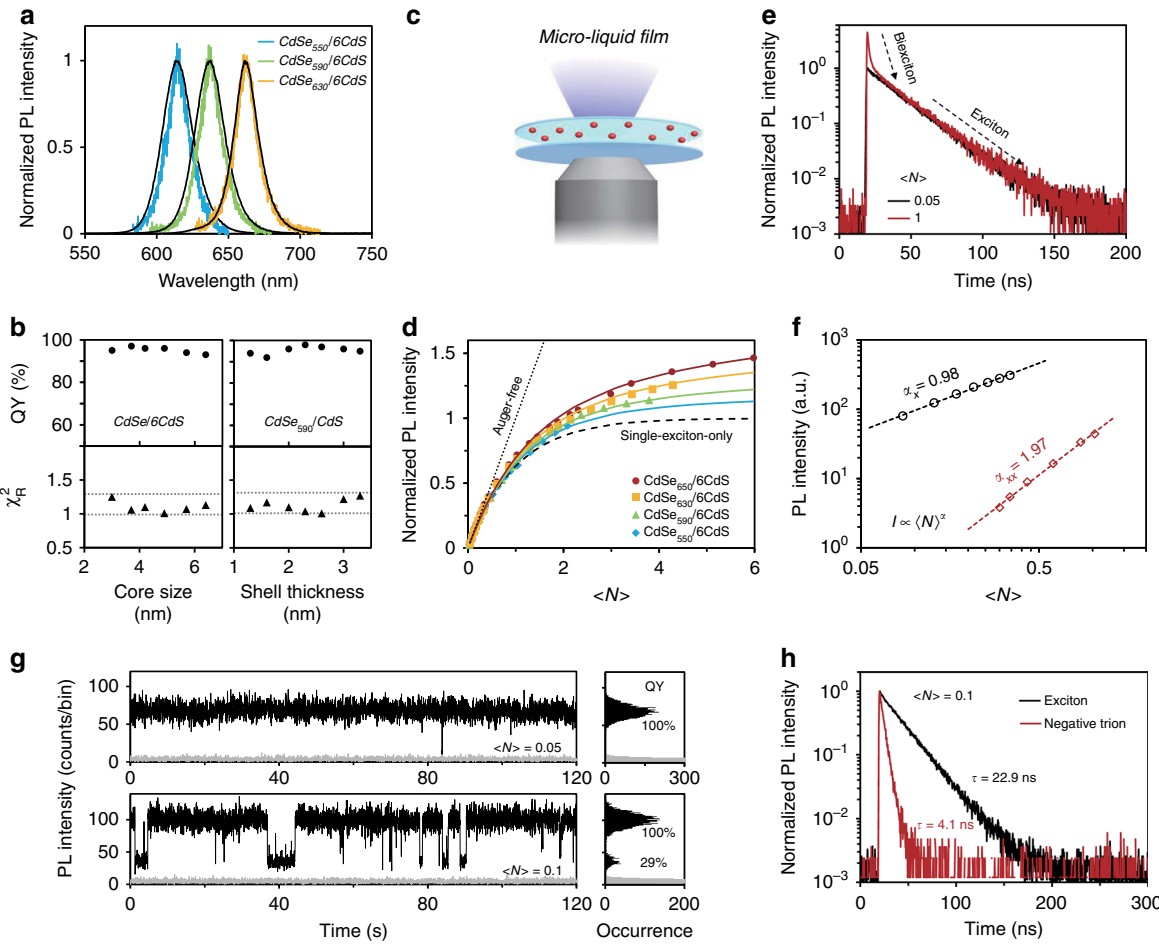

**Fig. 1** Optical properties of exciton, biexciton and trion for CdSe/CdS core/shell quantum dots (QDs). **a** Normalized photoluminescence (PL) spectra of representative single (color-coded) and the corresponding ensemble (black) QDs with different CdSe core sizes (the first-exciton absorption peak of CdSe core at 550, 590, and 630 nm, respectively) and six monolayers of CdS shell. **b** PL QYs and the goodness-of-fit ($\chi_R^2$) for mono-exponential fitting of PL decay curves for CdSe/CdS QDs with different core sizes or shell thicknesses. $\chi_R^2 = 1.30$ and 1.00 are represented by dotted lines. **c** Schematic of a micro-liquid film of QDs excited by a microscope objective. **d** Pump-power dependence of PL intensity for four representative QD samples with different core sizes but the same shell thickness measured by the micro-liquid film approach. Solid lines are the fits to the 'PL saturation'' model. The dashed and dotted lines are the PL saturation models with multi-exciton QY being 0% and 100%, respectively. **e** PL decay curves of CdSe₆₃₀/6CdS core/shell QDs as an representative sample with low (black) and relatively high (red) excitation power measured by the micro-liquid film approach. **f** Excitation power dependence of PL intensity of single-exciton (black circles) and biexciton (red diamonds). The slopes in the log–log plot for both channels match the expected values. **g** Representative PL intensity trajectories (black) of CdSe₆₃₀/6CdS core/shell QDs under different excitation power with background noises (gray) and the corresponding histograms. Bin time is 30 ms. **h** PL decay curves of the photons from the bright (black) and dim state (red) corresponding to **e** (<N> = 0.1)

relatively high biexciton QYs, the results are further verified by the second-order photon intensity correlation measurements of single QD (Supplementary Fig. 7).

Figure 1d shows that, as the core size increases, the saturation curves gradually depart from the single-exciton-only trend line—zero quantum yield for bi- and multi-exciton emission—and approach to the linear function (Auger-free line). This means that biexciton QY increases significantly by increasing the core size for the QDs with a fixed shell thickness (Supplementary Fig. 8). In sharp contrast, the biexciton emission QYs for QDs with a fixed core size but different shell thicknesses are almost constant (Supplementary Fig. 8), being consistent with the literature reports[29,41]. With $QY_{XX}$ and $k_{XX}$, the radiative and Auger non-radiative recombination rates of biexciton ($k_{rad,XX}$ and $k_{Auger,XX}$) can be calculated. $k_{rad,XX}$ shows similar dependence on core size and shell thickness, while $k_{Auger,XX}$ is much more sensitive to the core size than to the shell thickness (Supplementary Fig. 8 and Supplementary Note 5). Quantitative results in Supplementary

Fig. 8 suggest that, instead of increasing the shell thickness, varying the core size is a more efficient way to tune Auger recombination rate and a sole way to tune biexciton QY in CdSe/CdS core/shell QDs.

**Properties of negative trions.** Photoluminescence blinking of single QD allows one to study its trion states[14,25,42,43]. Single CdSe/CdS core/shell QD in this work turns from non-blinking to blinking by increasing the excitation power[12,14,44,45] (see Fig. 1g for example). As long as <N> is <0.5 (the linear range of saturation curves in Fig. 1d), the bright-state is equivalent to single-exciton emission and the photoluminescence QY remains near unity, and QY of the well-defined dim state in Fig. 1g should thus be 29%, accounting for identical absorption cross sections of bright and dim state at the short excitation wavelength (see detail below). In previous reports, with the bright-state being assigned to the neutral-state emission of single CdSe/CdS core/shell QD, the most commonly observed dim state under mild excitation

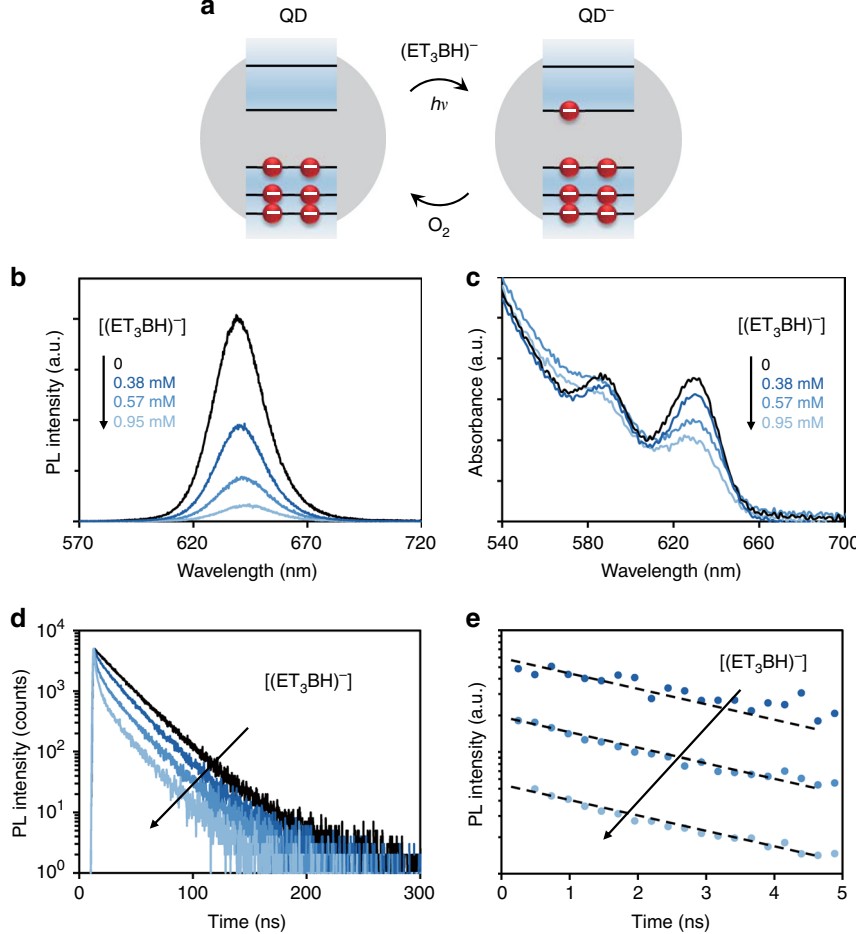

**Fig. 2** Photochemical electron-doping of CdSe/CdS core/shell quantum dots (QDs). **a** Schematic illustration of photochemical electron-doping and re-oxidizing processes. **b–d** Photoluminescence (PL) spectra, absorption spectra and PL decay curves of neutral (black) and negatively charged CdSe$_{590}$/8CdS core/shell QDs at various stages of photochemical doping (different shades of blue). **e** The extracted negative-trion decay dynamics at various stages of photochemical doping by subtraction of tail-normalized PL decay dynamics of neutral QDs from that of the negatively charged QDs. The black dashed lines are the mono-exponential fitting of the extracted decay dynamics

conditions is identified as a negative-trion emission state with different experimental methods[43,46,47]. This is further confirmed by photon statistics method (see detail discussion in Supplementary Note 8). While the photoluminescence decay dynamics of the bright-state in Fig. 1h is identical as the single-exciton decay dynamics, the mono-exponential decay dynamics of the dim state is assigned to the negative-trion emission decay dynamics (see detail below). Combining the emission QY and mono-exponential decay lifetime of the negative trion, one can readily determine its radiative and Auger non-radiative recombination rates $k_{\text{rad},X^-}$ and $k_{\text{Auger},X^-}$ (Supplementary Note 2).

To further verify the charge status and Auger recombination rate of the trion state, we apply photochemical doping in solution to directly prepare the negative trion state of colloidal QDs. It has been established that, upon photo-excitation of QDs, lithium triethylborohydride (LBH) in the solution can efficiently act as hole acceptor of QDs, which results in a negative trion in a QD[26,34,48] (Fig. 2a). Without efficient electron acceptors in the solution, the negatively charged QDs should be sufficiently stable for spectroscopic identification. As shown in Fig. 2b, by increasing the LBH concentration in the QD solution, the photoluminescence quantum yield of CdSe$_{590}$/8CdS QDs decreases. Simultaneously, the first-exciton absorption of the QDs is photo-chemically bleached without affecting their

**Table 1 Recombination rates of different channels for CdSe/CdS QDs determined by the photochemical doping method**

| QD sample | $k_X$ (ns$^{-1}$) | $k_{X^-}$ (ns$^{-1}$) | $k_{\text{Auger},X^-}$ (ns$^{-1}$) |
|---|---|---|---|
| CdSe$_{590}$/4CdS | 0.054 | 0.64 | 0.58 |
| CdSe$_{590}$/6CdS | 0.048 | 0.48 | 0.42 |
| CdSe$_{590}$/8CdS | 0.043 | 0.29 | 0.24 |
| CdSe$_{630}$/6CdS | 0.045 | 0.30 | 0.24 |

absorption at high energy (Fig. 2c and Supplementary Fig. 9). These results further confirms that there is an extra electron in the conduction band of CdSe/CdS core/shell QDs[49].

Because of the mono-exponential decay dynamics of single-exciton emission of the CdSe/CdS core/shell QDs used here, the photoluminescence decay rate of the photo-chemically doped QDs—QDs in their negative trion state—can be readily separated from that of the undoped QDs (Fig. 2d, e). Given the unity photoluminescence QY of the single-exciton emission, radiative and Auger recombination rates of the photo-chemically doped QDs at their trion state can be calculated from their photoluminescence QY and experimental decay rates (Table 1). Evidently, both radiative and Auger recombination rates determined by the photochemical doping approach are in good

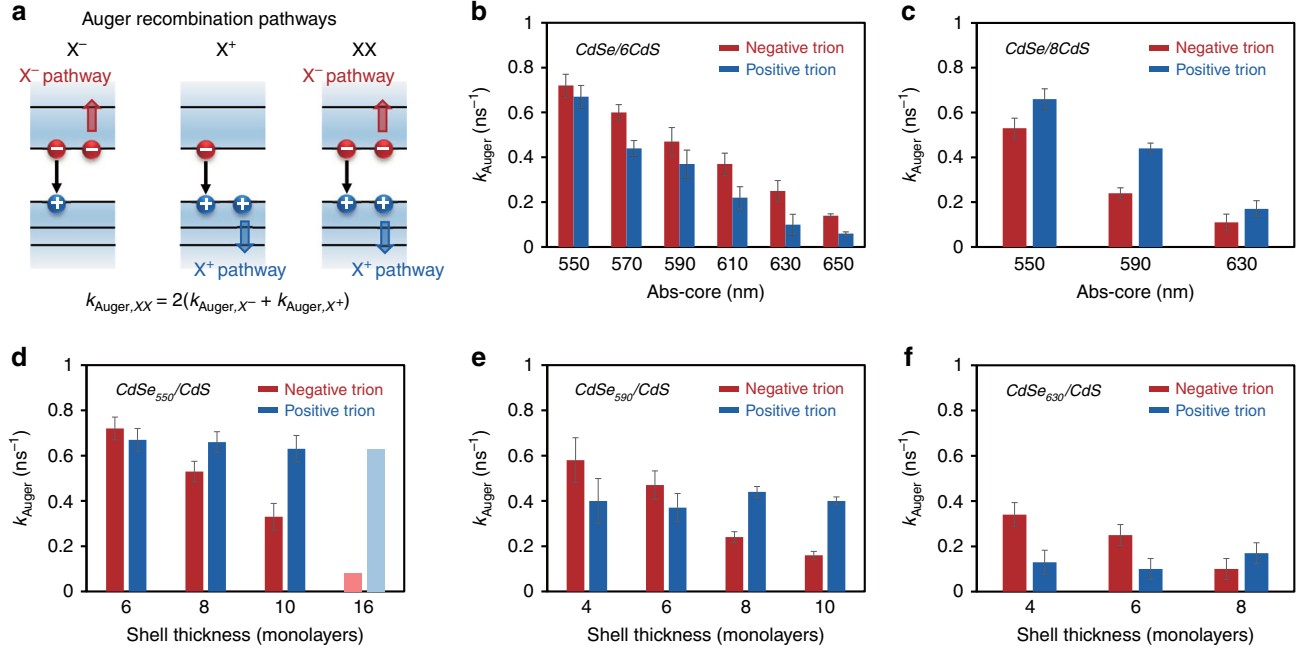

**Fig. 3** Core size and shell thickness dependence of Auger recombination rates. **a** Illustration of Auger recombination pathways for negative trion (X⁻), positive trion (X⁺), and biexciton (XX), and the relation among three rates. **b**, **c** Core size dependence of Auger recombination rates of X⁻ (red) and X⁺ (blue) for CdSe/CdS core/shell quantum dots (QDs) with four and eight monolayers of shells, respectively. **d–f** Shell thickness dependence of Auger recombination rates of X⁻ (red) and X⁺ (blue) for CdSe/CdS core/shell QDs with the first-exciton absorption peak of core at 550, 590, and 630 nm, respectively. The data for CdSe/CdS core/shell QDs with the first-exciton absorption peak of core at 550 nm and 16 monolayers of shell in **d** (light blue and light red) is from ref. [25]. Error bars are defined as s.d.

agreement with the rates determined by the single-dot spectroscopy for the CdSe/CdS core/shell QDs with various core size and/or shell thickness (Fig. 1h, Supplementary Table 3, and Supplementary Note 6), confirming the most prone to occur dim state observed by single-dot spectroscopy is a negative-trion emission state.

Importantly, photoluminescence, ultraviolet (UV)–visible (Vis) absorption, and transient photoluminescence spectral properties are reversible upon exposure of the doped QDs to oxygen (Supplementary Fig. 9). This indicates that the spectral bleaching and appearance of the short-lifetime photoluminescence decay channel (Fig. 2) are not due to surface degradation of the QDs caused by the chemical reduction. Recovery by exposure to oxygen—a common electron acceptor[45]—is also consistent with negatively charging nature of the doped QDs.

With the Auger recombination rates for both biexciton and negative trion, one could further obtain Auger non-radiative recombination rate of the corresponding positive trion ($k_{Auger,X^+}$) using the superposition principle (Fig. 3a, Supplementary Fig. 10, and Supplementary Note 7)[5,25,26]. For CdSe/CdS core/shell QDs, the superposition principle, namely $k_{Auger,XX} = 2\left(k_{Auger,X^+} + k_{Auger,X^-}\right)$, has been repeatedly verified in both single-dot[25] and ensemble[26] experiments and widely applied in theoretical studies[50] as well. For the current system, our measurements can directly determine the three rates ($k_{Auger,XX}$, $k_{Auger,X^+}$, and $k_{Auger,X^-}$) (see below) for some samples, which also confirms the superposition principle (Supplementary Table 6).

**Crossover of Auger rates of positive and negative trions**. Figure 3b illustrates Auger recombination rates of negative and positive trions of CdSe/6CdS core/shell QDs with different core sizes. In sharp contrast to literature[25,26,31], the Auger rates of

negative trions of these QDs are universally faster than those of the corresponding positive trions. As core size increases in the experimental range, ratio between Auger rates of positive and negative trions reduces from ~90% to ~45%. Interestingly, for QDs with 8 monolayers of the CdS shells, Fig. 3c gives results qualitatively similar to that reported in literature[25,26,31], i.e., the Auger recombination rates of negative trions being always slower than those of the corresponding positive trions. Figures 3b, c suggest that a crossover of the Auger recombination rates of negative and positive trions should exist for the QDs with any given core size, but different shell thicknesses. This is confirmed by systematic measurements for QDs with different core sizes (Fig. 3d–f). It should be noted that, with either hexahedral or spherical shape of the core/shell QDs[23], the trends shown in Fig. 3 are almost identical (Supplementary Fig. 11).

Literature[3,4,34] reported that Auger recombination lifetime scales linearly with the nanocrystal volume (known as "volume scaling"). However, Fig. 3d–f demonstrate that, the rates of the two basic types of Auger recombination evolve in different patterns with their shell thicknesses (or total volume). Quantitatively, for the positive trion, Auger recombination rate is in good volume scaling with the core of the core/shell QD ($R_{core}^{2.8}$, $R_{core}$ as the core radius of the core/shell QDs) but exhibits poor correlation with the whole volume of core/shell QD (Supplementary Fig. 12). Conversely, Auger recombination rate of the negative trion is in good volume scaling with the whole core/shell QD ($R^{3.7}$, $R$ as the core/shell total radius) but not well correlated with the core radius (Supplementary Fig. 12). Overall, doubling the core radius (or whole radius) of a core/shell QD increases the Auger recombination lifetime of positive (or negative) trion by around one order of magnitude.

It should be noted that, majority of studies related to Auger recombination on CdSe/CdS core/shell QDs have been focused on small cores (typically ~3 nm in size) and thick CdS shells

(>10 monolayers)[25,26,42], which should thus not observe the rate crossover discussed above. For example, one set of experimental results from literature[25] is plotted along with the results in this work (Fig. 3d). Evidently, though the literature results were obtained using a different method, they were consistent with the general geometry-dependent trends for both types of Auger recombination. In addition, recent reports[36] discovered that the relatively poor optical quality of the CdSe/CdS core/shell QDs studied previously might have caused large uncertainty to resolve the crossover.

**Inversion of quantum yields of negative and positive trions**. The crossover of Auger rates between the two basic types of Auger channels should result in inversion of their QYs for the CdSe/CdS core/shell QDs. Because of the comparatively low Auger rates of the QDs with large core sizes, their crossover of Auger recombination rates and inversion of emission QYs of positive and negative trions should be readily observable in single QD experiments.

Figures 4a, b, respectively, show the photoluminescence intensity trajectories for representative single QDs for CdSe$_{630}$/8CdS and CdSe$_{630}$/4CdS core/shell samples. For the top trajectory in each case, excitation power is increased to a medium level to induce a decent amount of the first dim state (noted as 'Dim 1'). By further increasing the excitation power (still satisfying $<N>$ <0.5 to guarantee the bright-state being mainly single-exciton emission), an additional dim state (noted as 'Dim 2') becomes visible in the bottom trajectory for each case. For the CdSe$_{630}$/8CdS core/shell QD in Fig. 4a, the emission QYs of the first and second dim states are ~46% and ~31%, respectively. Conversely, the emission QY of the first dim state ('Dim 1', 20%) is significantly lower than that of the second one ('Dim 2', ~35%) for

the CdSe$_{630}$/4CdS core/shell QD in Fig. 4b. Figures 4c, d reveal that, for either of the two QDs, the mono-exponential lifetime values of photoluminescence decay dynamics of both bright-state and 'Dim 1'—unaltered by changing the excitation power—are consistent with the single-exciton and negative trion emission, respectively. Similarly, 'Dim 2' for both QDs possesses mono-exponential photoluminescence decay dynamics (Fig. 4c, d), implying 'Dim 2' is unlikely a mixture of multiple emissive states.

Figure 4e illustrates that the inversion of emission QYs of 'Dim 1' and 'Dim 2' is statistically reproducible, given the diagonal line being equal QYs for the two dim states. With both photoluminescence quantum yield and mono-exponential decay lifetime of 'Dim 1' and 'Dim 2', one can directly calculate their Auger recombination rates (Fig. 4f). As demonstrated above, the most prone to occur dim state ('Dim 1') for single CdSe/CdS core/ shell QD is confirmed to be the negative-trion emission[43,46,47]. Nature of the second occurring dim state ('Dim 2') needs to be determined experimentally, which could be either positive trion or negative tetron (three electrons and one hole) after excluding the possibility of being a mixed state.

For Auger recombination, theoretical[4,47] and experimental[51] studies reported that Auger recombination rate of a negative tetron should be 3–5 times faster than that for the corresponding negative trion, which is inconsistent with the results for multiple dots for either sample (Fig. 4f). In fact, instead of being 3–5 times faster than the negative trion, Auger recombination rate of 'Dim 2' for CdSe$_{630}$/4CdS core/shell QDs is statistically below 30% of that of the corresponding negative trion (Fig. 4f). Thus, results in Fig. 4f confirm that 'Dim 2' cannot be associated with negative tetron. Additional measurements (Supplementary Figs. 13 and 14, Supplementary Table 4, 5 and 6, and Supplementary Note 8) further support that 'Dim 1' and 'Dim 2' in Fig. 4 are originated from the emission of negative and positive trions, respectively.

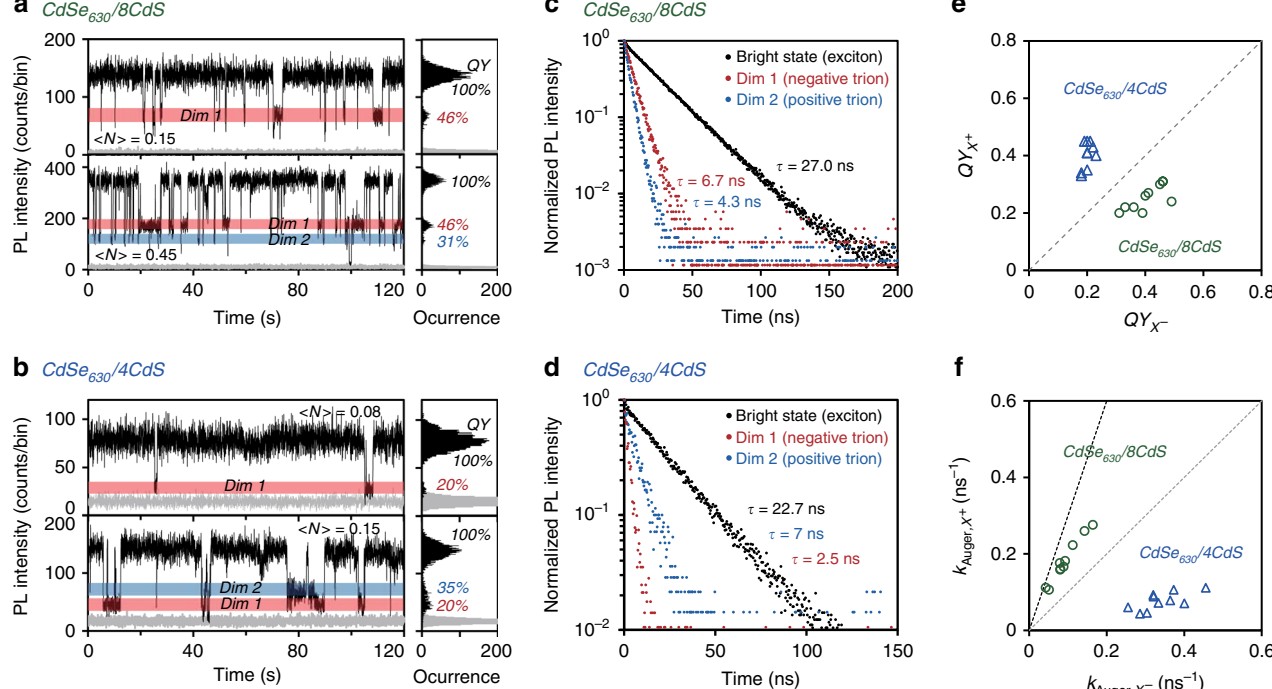

**Fig. 4** Inversion of emission quantum efficiency of negative and positive trions. **a**, **b** PL intensity trajectories and histograms for single CdSe$_{630}$/8CdS and CdSe$_{630}$/4CdS core/shell quantum dots (QDs). The two distinct dim states with different emission intensities are shaded in red ('Dim 1') and blue ('Dim 2'), which are corresponding to negative and positive trion states, respectively. **c**, **d** PL decay curves of the three emission states in **a** and **b**. **e**, **f** The plot of quantum yields (QYs) (**e**) and Auger rates (**f**) of positive trions versus those of negative trions for ten single CdSe630/8CdS (green circles) and CdSe630/4CdS (blue triangles) core/shell QDs . The dashed line (gray) in **e** represents $QY_{X+} = QY_{X-}$. The gray and black dashed lines in **f** represent $k_{Auger,X+} = k_{Auger,X-}$ and $k_{Auger,X+} = 3k_{Auger,X-}$, respectively

The diagonal line in Fig. 4f indicates equal Auger rates of positive and negative trions. The distribution of the experimental results in Fig. 4f statistically confirms both key findings in the above section, i.e., a slower Auger recombination rate for the negative trion than the corresponding positive trion for QDs with specific geometric structures and a rate crossover between the two basic Auger recombination channels by varying the geometric structure of the core/shell QDs. Furthermore, these results are found to be consistent with those based on the superposition principle (Supplementary Table 6).

In the literature, positive trion of single CdSe/CdS core/shell QD has been reported to possess a lower emission QY than the corresponding negative trion does[25,26]. However, Fig. 4b, e indicate that the emission QY of a positive trion can be much higher than that of the corresponding negative trion. As expected, the inversion of the emission quantum yield (Fig. 4e) is correlated with the rate crossover of the two basic channels of Auger recombination (Fig. 4f). These results demonstrate that Auger engineering through varying geometry of the core/shell QDs can specifically manipulate optical and optoelectronic properties of multi-carrier states of QDs, such as the blinking behavior of single QDs.

**Theoretical interpretation**. The electronic structures of CdSe/CdS core/shell nanocrystals have been studied intensively. It is well established that the valence band maximum (VBM) state is strongly confined in the CdSe core as a result of the large valence band offset between CdS and CdSe[52,53]. However, their conduction band offset is small[54], thus the conduction band minimum (CBM) confinement in the core/shell structure is much weaker[55]. Depending on the shape and size of the core and shell, the band alignment can be either type-I or quasi-type-II[52,53,55]. Theoretical studies also revealed that the Auger processes for positive and negative trions in CdSe/CdS QDs are asymmetric. From tight-binding calculations, Sargent and colleagues[50] reported that in a CdSe/CdS QD with 4 nm core and 10 nm shell, Auger recombination can be six times faster for positive trions compared to negative ones. Using the k·p method, Efros et al. showed that increasing the CdS shell thickness can lead to a much stronger suppression on the Auger recombination for negative trion than for positive trion[56]. These calculations agree with some of our experimental results.

In order to thoroughly understand the unexpected experimental phenomena descried above, time-dependent perturbation theory[5] coupled with density functional theory with the charge patching method[57] and the folded spectrum method[58] is applied for theoretical study of the Auger recombination in CdSe/CdS core/shell QDs (see Supplementary Note 9). Near the band-edge of a CdSe/CdS core/shell QD, the density of hole energy levels is much higher than that of electron energy levels, which suggests more Auger recombination pathways for a positive trion hence more efficient Auger recombination for positive trion than negative trion[50,56]. However, the picture is incomplete without considering the coupling between initial and final states for each pathway, where geometry-dependent dielectric screening should play a critical role.

Using Slater determinants to describe the initial and final many-particle states in Auger recombination, we can evaluate their coupling strength based on Coulomb integrals in the form

$$J(j,k,l,m) = \iint \left[ \varphi_j^*(\mathbf{r})\varphi_k^*(\mathbf{r}')\varphi_l(\mathbf{r})\varphi_m(\mathbf{r}')e^2 \right] / [\epsilon(\mathbf{r},\mathbf{r}')|\mathbf{r}-\mathbf{r}'|]d\mathbf{r}d\mathbf{r}'$$

,here $\{\varphi\}$ are the wavefunctions of the relevant single-particle states[5]. When the initial single-particle states are degenerate or nearly degenerate, a configuration-interaction (CI) expansion of the many-particle states is used to account for the coupling between the nearly degenerate Slater determinants[59]. This is

especially important for the positive trion due to the almost degenerate single-particle states.

The term $\epsilon(\mathbf{r},\mathbf{r}')$ is dielectric function accounting for the screening effect. Larger $J$ leads to higher Auger rate. Actually, the dielectric constant can change with the quantum dot size[60]. Theoretically, one approach is to describe the reduction of the overall dielectric constant as a consequence of the increase band gap in the QD. For example, Efros et. al. investigated the intraband and interband Auger processes in CdSe/CdS core/shell QDs by introducing reduced dielectric screening as an input parameter in the calculation. The best agreement[61] between theory and experiment was obtained with the effective dielectric constant being ~ 40% of the bulk semiconductor. Alternatively, the reduced overall dielectric constant can be described as a reduction of dielectric response for the locations near the inorganic–organic interface[62]. Such microscropic picture of the dielectric response allows us to develop a location dependent dielectric function model, which is used in the current work.

For Auger recombination of a negative trion, the relevant Coulomb integral is $J(e^1, e^2, e_n, h)$, where $e^1$, $e^2$, and $h$ are the two CBM states and one VBM state involved in the trion, and $e_n$ is the final electron state in the higher conduction band. The corresponding Auger process is: $e^2$ recombines with $h$, and $e^1$ jumps into $e_n$. Similarly, the relevant Coulomb integral for Auger recombination of a positive trion is $J(h^1, h^2, h_n, e)$, corresponding to $h^2$ recombines with $e$, and $h^1$ jumps to $h_n$. Thus, the magnitude of either $J$ strongly depends on the characters of CBM and VBM states involved in a specific Auger recombination. From this aspect, co-localization of all wavefunctions within the geometric boundary of a QD will lead to efficient Auger recombination for either positive or negative trion in comparison to the corresponding bulk crystal. The small geometric size of a QD would also enable significant penetration of electric-field lines of the carriers into the ligand/solvent environment, whose very-low dielectric constants would largely reduce the dielectric screening of those wavefunctions close to the interface and hence accelerate Auger recombination of the corresponding trion.

For three representative QDs, Fig. 5a illustrates characters of the relevant states involved in Auger recombination, namely, CBM, VBM, representative high-energy electron ($e_n$), and high-energy hole ($h_n$) states. The valence band offset is much larger than the conduction band offset between CdSe and CdS and the electron is lighter than the hole[52], which implies that CBM wavefunction is much more delocalized than the corresponding VBM wavefunction in CdSe/CdS core/shell QDs. The high-energy $e_n$ and $h_n$ states are delocalized over the whole QD.

When the shell is thin, wavefunction of CBM extends into regions close to the inorganic–organic interface, where dielectric screening is much weaker than the QD interior. This leads to larger Coulomb integral $J$ for a negative trion, thus higher Auger rate. Given a fixed core size, increase of the CdS shell thickness pushes CBM wavefunction away from the interface (Fig. 5a), implying increased dielectric screening for the negative trion. Simultaneously, CBM wavefunction is also more delocalized, which decreases the $e^1/e_n$ overlap. Both factors would contribute decrease of the Coulomb integral $J$ and result in a reduced Auger rate of the negative trion. However, increasing shell thickness has small effects on the $h^1/h_n$ overlap and the dielectric screening, implying insensitive Auger rate of positive trion to varying the shell thickness. These combined effects on negative and positive trions would result in the Auger rate crossover observed in Fig. 3d–f, which is captured by our computation using the model discussed above (Fig. 5b).

When the core size of core/shell QDs increases with a fixed shell thickness, wavefunctions of both CBM and VBM are

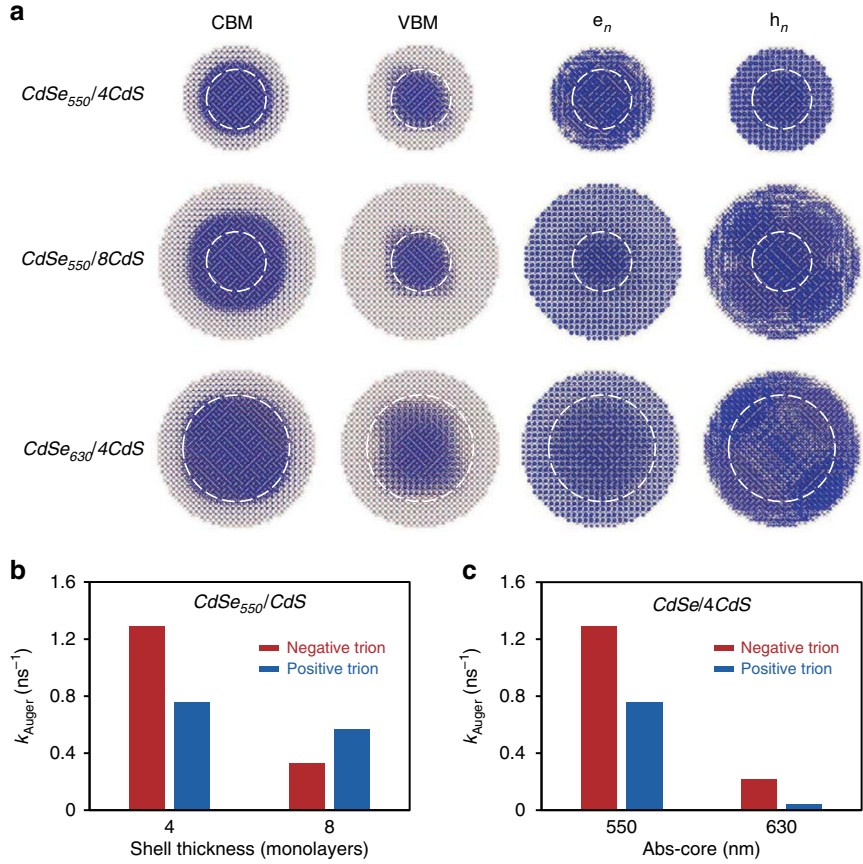

**Fig. 5** Snapshot of the charge density and the theoretical calculation results. **a** The charge density of the conduction band minimum (CBM), valence band maximum (VBM), and averaged $e_n$ and $h_n$ states for $CdSe_{550}/4CdS$ quantum dots (QDs), $CdSe_{550}/8CdS$ QDs and $CdSe_{630}/4CdS$ QDs. **b** The theoretical calculation results of the Auger recombination rates for negative and positive trions for CdSe/CdS core/shell QDs with the same core size but different shell thicknesses. **c** The theoretical calculation results for the Auger recombination rates for negative and positive trions for CdSe/CdS core/shell QDs with the same shell thickness but different core sizes

confined more in the core but become more delocalized within the core. These factors cause significant decrease of the $h^1/h_n$ and $e^1/e_n$ overlap and enhanced dielectric screening for both CBM and VBM, which results in simultaneous decrease of Auger rates for both types of trions (Fig. 3b, c). Given the relatively small radius of wavefuntion of VBM than that of CBM (Fig. 5a), the effects of the core size would have more impact on the positive trion than on the negative one. Again, our computation captures the trends (Fig. 5c).

## Discussion

For commonly studied II−VI and III−V semiconductors, effective mass of hole is significantly heavier than that of the corresponding electron[63]. As a result, for type-I—bottom of conduction band (top of valence band) of the core being lower (higher) than those of the shell—core/shell QDs, the band-edge hole wavefunction (VBM in Fig. 5) is more localized within the inner part of a core/shell QD than the band-edge electron wavefunction (CBM in Fig. 5)[4,50]. Thus, the two wavefunctions could experience very different dielectric screening, given the ligand/solvent environment possessing much smaller dielectric constant than the semiconductors. Alternatively, we can state that the dielectric screening becomes geometry-dependent for core/shell QDs and it is no longer suited to treat dielectric screening as the same constant for the two types of basic Auger channels in a given core/shell QD[5,50,64]. Specifically, for certain core/shell geometry,

positive trion may experience substantially stronger dielectric screening than the corresponding negative trion, which would cause reduced Auger rates for the positive trion in comparison with the corresponding negative trion. Evidently, this is a counter factor to the other well-known factors, such as density of states and two types of wavefunction overlapping discussed above. As pointed out above, the latter ones always suggest more efficient Auger recombination of positive trions than that of the corresponding negative trions for CdSe/CdS core/shell QDs[4,25,26,50,56].

Crossover of the Auger rates of two types of basic channels observed here has neither been predicted by theory nor reported experimentally yet. The experimental and theoretical results suggest that, for the CdSe/CdS core/shell QDs—regardless of the core size—with relatively thin CdS shells, the counter effect of geometry-dependent screening can dominate the other factors for determining the relative Auger rates of the two basic channels. This would result in overall less efficient Auger recombination of positive trion than that of the corresponding negative trion for those specific QDs (Fig. 3). Figures 3b and 5c show that, as the core size increases, the dominating role of geometry-dependent dielectric screening for the QDs with relatively thin CdS shells becomes more dramatic. As the shell thickness increases, the other factors well documented in literature overtake the dominating role and cause rate crossover between the Auger recombination of two basic channels.

Geometry-dependent dielectric screening not only is responsible for the Auger rate crossover but also represents a new

strategy for Auger engineering, which can independently tune the rate of either basic type of Auger recombination. For instance, by coating core/shell QDs with a medium with high (or low) dielectric constant—either inorganic shell or organic solvent/matrix, Auger recombination rates can be greatly reduced (or enhanced). We are actively working on these directions, aiming to further enlarge the tuning range of the rates of two basic types of Auger recombination.

In terms of non-radiative carrier dynamics, two elementary processes are important. One is the carrier cooling assisted by electron-phonon coupling after excitation, and the other is the electron-electron interaction induced Auger recombination. In the current experiment, the measured photoluminescence decay dynamics is in the nanosecond time range after the initial excitation. This is far beyond the initial hot carrier cooling, which typically happens within a few picoseconds. We can thus assume the Auger processes start from the ground state of a trion.

There is a phonon-assisted Auger process[65,66]. Within this process, not only the electron (or hole) can be excited to their higher energy states through the electron-electron interaction, but phonons can be absorbed or emitted. However, these additional channels do not change the overall oscillator strength of the Auger process. Instead, they just spread out the original zero phonon Auger line, thus broadening the Auger peak. In the current study, this effect has been approximated by using a finite broadening of the energy delta function in the Fermi golden rule in Eq. S6. We have used a peak width of 10 meV (close to kT) to represent this effect. Further increasing the peak width does not significantly change the results.

In summary, our results suggest that further suppression (or enhancement) of Auger recombination is feasible with the new strategy. Related, improvement (or reduction) of the quantum yields of multi-carrier states—bi/multi-excitons and trions—of QDs should be possible. Although the results here are obtained with CdSe/CdS core/shell QDs, above discussion tells us that the new strategy based on geometry-dependent dielectric screening should be generally applicable to nearly all types of core/shell QDs. The core/shell QDs with rationally engineered Auger recombination should greatly promote QDs for important applications, such as high-power light-emitting-diodes, lasers, single-molecule tracking, super-resolution microscopy, and advanced quantum light sources.

## Methods

**Sample preparations**. The zinc-blende CdSe/CdS core/shell QDs with different core sizes and shell thicknesses were prepared according to the method reported in our previous report[23]. For conventional ensemble optical measurements, QD samples were diluted in toluene until the optical density was around 0.1 at the first-exciton absorption peak. QD micro-liquid films were fabricated by transferring a droplet of QD solution (~ 10 μL) to a clean cover slip and mounting another cover slip on the top. The thickness of QD liquid film between two cover slips was around 10 μm determined by imaging the uppermost and lowermost emitting QDs in the liquid film using a fluorescence microscope (see details below). For single-dot spectroscopy experiments, diluted QD solution in toluene with 2.5% PMMA by weight was spin-casted on a clean cover slip. All optical measurements were performed at room temperature.

**Optical measurements on QDs at ensemble level**. UV−Vis spectra were taken on an Agilent Technologies Cary 4000 UV−visible spectrophotometer. Steady-state and transient photoluminescence spectra were recorded on an Edinburgh Instruments FLS920 spectrometer. Transient photoluminescence spectra were measured via the time-correlated single-photon counting (TCSPC) method with a 405 nm picosecond laser diode with 2 MHz repetition rate. The photoluminescence lifetime $\tau$ was obtained by fitting the decay curves by a single-exponential decay function with an acceptable goodness-of-fit ($\chi_R^2$). The absolute photoluminescence QY was measured using an Ocean Optics FOIS-1 integrating sphere coupled with a QE pro spectrometer. The system was calibrated with a DH-3-cal standard light source.

**Optical measurements on QDs using fluorescence microscope**. Optical properties of QD micro-liquid films and single QDs were measured using an inverted epifluorescence microscope (Olympus IX 83) equipped with a 60 × oil immersion objective (N.A. = 1.49) and suitable spectral filters. In order to measure the steady-state photoluminescence spectra of single QDs, a 405 nm continuous wave laser (PicoQuant) was used as the excitation light source and a spectrometer with EMCCD (Andor Shamrock 303i) was used to record the spectra.

The 405 nm laser was switched to a pulsed mode with 1 MHz repetition and ~ 50 ps pulse width for time-resolved photoluminescence spectra measurements. For measuring the biexciton emission lifetime, a 405 nm femtosecond pulsed laser (Light Conversion Pharos + Orpheus + LYRA) with 1 MHz was used instead.

In order to measure the biexciton lifetime and QY of QD micro-liquid films, the emission signals were collected and recorded by a single-photon counting system (a Micro Photon Devices MPD-050-CTD-FC photon detectors with 50 ps resolution and PicoQunat Picoharp 300 TCSPC module) in a TCSPC mode. With different excitation power densities, a series of transient photoluminescence spectra were obtained and thus the biexciton lifetime and QY could be calculated.

Photoluminescence intensity trajectories and transient photoluminescence spectra of single QD were measured simultaneously with the same single-photon counting system mentioned above in a time-tagged time-resolved (TTTR) mode. The integration time for each data point in the photoluminescence intensity trajectories was 30 ms. Second-order photon intensity correlation measurements were performed with the same system and module with a Hanbury-Brown and Twiss (HBT) intensity correlation set-up comprised of a 50/50 beam splitter and two single-photon detectors.

**Photochemical electron-doping of QDs**. Lithium triethylborohydride (Li(Et₃BH), 20 μL of 1 mol/L) in tetrahydrofuran (Sigma-Aldrich) was added to 1 mL of anhydrous toluene to form photochemical electron-doping solution. The solution of CdSe/CdS core/shell QDs (optical density of band-edge absorption peak < 0.1) in anhydrous toluene and 10 μL of trioctylphosphine was loaded into an air-tight cuvette. Into the QD solution in the cuvette, the Li(Et₃BH) solution was added stepwise with 50−1000 equivalent of CdSe/CdS QDs. The resulting solutions were exposed to a 365 nm LED (~ 6 mW) for photochemical electron-doping. After a few minutes of photo-radiation, the QD sample reached a steady-state, indicated by stable absorption and photoluminescence spectra.

All above operations were performed at room temperature and under N₂ atmosphere in a glove box. By varying the amount of Li(Et₃BH) added, the portion of negatively charged QDs in a solution was controlled.

## Data availability

The authors declare that all data supporting the findings of this investigation are available within the article, its Supplementary Information, and from the corresponding authors upon reasonable request.

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

## Acknowledgements

This work was supported by the National Program on Key Research and Development Project (2016YFB0401600) and the National Natural Science Foundation of China (Grants 21573194, 21803053 and 21703202). J. Kang and L.W. Wang were supported by the Director, Office of Science, the Office of Basic Energy Sciences (BES), Materials Sciences and Engineering (MSE) Division of the U.S. Department of Energy (DOE) through the organic/inorganic nanocomposite program (KC3104) under contract DE-AC02-05CH11231. It used the computational resource of the Oak Ridge Leadership Computing Facility at the Oak Ridge National Laboratory through the INCITE project.

## Author contributions

H.Q. and X.P. conceived the idea and supervised the experimental studies. L.W.-W. designed and supervised the theoretical studies. X.H. conducted the synthesis and spectroscopy experiments. J.K. performed the theoretical calculations. J.M. designed the micro-liquid film experiments. J.Z. supported synthesis. X.C., L.C., and L.W. participated in the interpretation of results.

## Additional information

**Competing interests:** The authors declare no competing interests.

