## [Peer Review File · Nature Communications]

Reviewers' comments:

Reviewer #1 (Remarks to the Author):

This manuscript aims at developing ways to engineer Auger recombination in CdSe/CdS core shell nanocrystals. The main finding is the claimed different evolution of positive and negative trion recombination rate compared to literature. In general, the goal is important and ambitious. However, it was not clear to me how the positive and negative trions and bi-excitons were identified. In Fig.1a we see emission lines from single and multiple qdots with different size. There is no sign of trion or XX as is commonly observed in self-assembled qdots or 2D materials. The presence of XX is indirectly deduced from the intensity dependence and decay of PL, figs 1c,d. The central object of this investigation, trion, is deduced from blinking of PL. The authors claim, based on literature, that they observe blinking involving a negative trion and positive trion is inferred indirectly. In other semiconductors, e.g., self-assembled qdots, dots are loaded with positive or negative charges with a gate. What I see here is all indirect observation, yet conclusions are drawn about positive and negative trion Auger rates. The authors claim that they engineer Auger rates by engineering screening with different ration of core to shell. The screened Coulomb interaction is the same for positive and negative trions, I fail to understand how screening differentiates between the two trions and can lead to engineering Auger rates. The Authors evaluate Auger rate in lowest order of Coulomb interaction as done in a number of papers. What differentiates negative and positive trions are the wavefunctions, not screening, which the authors compute in a way similar to their previous work, Refs.4,49. The problem of Auger processes is quite a bit more complicated beyond lowest order perturbation theory as illustrated for CdSe qdots by Voznyy and co-workers in Phys.Rev.B 84, 155327 (2011) and I would encourage authors to pursue this research. Perhaps I missed it but if the authors could unambiguously identify charge of a trion,e.g., with a gate or some other way? and show engineering of Auger rates with different surrounding screening environment, I would be happy to reconsider this paper for publication.

Reviewer #2 (Remarks to the Author):

The work is scientifically sound good. The problem is worth of investigation, due to the experimental and especially theoretical motivations. In this paper an experimental investigation have been demonstrated and agreed with the theoretical studies, but there are a lot of interesting results and comments on them in theory. Related literature is reviewed and obtained results have been compared with it.

Despite the above, the authors should clarify some key aspects for the publication of the manuscript, which in my opinion are mandatory.

The following observations should be analyzed carefully and keep in mind the respective changes:

1. I recommend to include other references for theoretical studies of CdSe/CdS core shell QDs.
2. In the manuscript the authors analyze the experimental properties of CdSe/CdS core shell QDs that deals with the theoretical studies. What is missing is a description of results in the literature on different geometries depend dielectric screening that were already studies and a comparison with the observed results.
3. A phonon carrier studies is recommended in this paper, the study of exciton properties were done - what is missing is a description on the phonons interactions, the interface on phonons play an increasingly large role to modify the size in QDs, so I suggest to add a paragraph about the phonon carrier interactions for the studied QDs.

Reviewer #3 (Remarks to the Author):

This manuscript reports measurements of Auger recombination rates in core/shell quantum dots with different core and shell sizes. Excellent results are obtained through a combination of synthesis of samples with outstanding quality and careful, thorough measurements. Biexciton, negative-trion, and positive-trion Auger rates are separately determined, and the novel observation is made that the positive- and negative-trion rates depend differently on shell thickness, with particular geometries enabling a reversal of the usual ratio of Auger rates. The explanation in terms of dielectric screening is convincing, has to the best of my knowledge not been properly considered before, and provides a new tool to enable control over Auger rates. I recommend publication of the manuscript in Nature Communications.

The only significant comment that I have on the manuscript is that quality of written English is poor in several places. The manuscript should be carefully proofread by a native English speaker or a professional editing service before it can be published.

Responses to the comments from Reviewer #1.

Comments: This manuscript aims at developing ways to engineer Auger recombination in CdSe/CdS core shell nanocrystals. The main finding is the claimed different evolution of positive and negative trion recombination rate compared to literature. In general, the goal is important and ambitious.

Response: Thanks for the encouragement. No action is needed.

Comments: However, it was not clear to me how the positive and negative trions and bi-excitons were identified. In Fig.1a we see emission lines from single and multiple qdots with different size. There is no sign of trion or XX as is commonly observed in self-assembled qdots or 2D materials. The presence of XX is indirectly deduced from the intensity dependence and decay of PL, figs 1c,d.

Response: Thanks for your suggestions and we have addressed the state assignment issue in the revised manuscript. Before going to details, it should be a good idea to state clearly that colloidal quantum dots are different from self-assembled quantum dots in their targeted temperatures for applications and studies. To exploit the outstanding optical properties of colloidal quantum dots at room temperatures, this and many other studies focus their properties under ambient conditions. As a result, it is expected not to observe emission peaks of all multi-carrier states in Fig. 1a, which are recorded at room temperature with moderate excitation power.

For the main theme in this work, we agree that identification of biexciton and negative trion provides necessary foundation. Following the suggestion of the referee, we provide solid evidences to identify both types of multi-carrier states using well-established methods.

In the revised manuscript, we add the excitation power dependence of emission intensity of biexciton (Fig. 1d inset and Fig. S5 in the Supplementary Information). As shown below in Fig. R1, The emission intensity for each state is extracted by bi-exponential fitting of the emission decay curves with different excitation power, i.e. different $\langle N \rangle$. One can find that the emission intensity of the component assigned to single-exciton (X) is linearly dependent on $\langle N \rangle$, whereas the emission intensity of the component assigned to biexciton (XX) follows quadratic power law.

Figure R1. Excitation-power dependence of emission intensity of single-exciton (black circles) and biexciton (red diamonds) for **a.** CdSe₆₃₀/6CdS QDs and **b.** CdSe₆₅₀/6CdS QDs. The slopes in the log-log plots for all channels match the expected values.

For identification of the negative trion, the original submission offered a detailed account for our system using spectroscopic methods published in Ref 47 (Xu Nano Lett 17, 7487, 2017) combining second-order photon correlation and time-resolved spectroscopy (see Supplementary Information 14). In this revision, we adopt an established charge injection method to further confirm the sign of charge (see response to the next comment below).

Comments: The central object of this investigation, trion, is deduced from blinking of PL. The authors claim, based on literature, that they observe blinking involving a negative trion and positive trion is inferred indirectly. In other semiconductors, e.g., self-assembled qdots, dots are loaded with positive or negative charges with a gate. What I see here is all indirect observation, yet conclusions are drawn about positive and negative trion Auger rates.

Response: Yes, trion—especially negative trion—is the central object of this investigation. We agree that it is a good idea to directly inject a charge with definite sign into a quantum dot to identify the trion. Different from self-assembled quantum dots on solid substrate, injection of a charge into a colloidal quantum dot is easier to realize in solution through chemical means.

Inspired by the suggestion of this referee, we have examined all existing methods for direct injection of a charge with definite sign into a quantum dot. Among them, the best performed one is injection of electrons into CdSe (or its core/shell) quantum dots reported by two respectable groups (Rinehart J. Am. Chem. Soc 135, 19782, 2013 and Wu ACS Nano 11, 8437, 2017), which yields negative-trion state in a quantum dot. With significantly higher optical quality of our CdSe/CdS core/shell quantum dots, this well-established protocol yields definite identification of negative-trion state for our samples. The results are included in Fig. 2 in the revised text and Fig. S9 in the revised Supplementary Information. The corresponding experimental details are given in the Method Section. In short, the Auger rates of negative trions with different core size and/or shell thickness

determined by this photochemical doping method well agree with the results obtained using the method outlined in Figure 1.

Similar to what reported in literature, direction injection of a hole into a quantum dot in solution (Wu ACS Nano 11, 8437, 2017) is found to be irreversible and thus less reliable. Because positive-trion state is not the key element in the current scheme (Auger rates of positive trions being determined by bi-exciton and negative trions), we decide not to include this part in the revised manuscript.

Comments: The authors claim that they engineer Auger rates by engineering screening with different ration of core to shell. The screened Coulomb interaction is the same for positive and negative trions, I fail to understand how screening differentiates between the two trions and can lead to engineering Auger rates. The Authors evaluate Auger rate in lowest order of Coulomb interaction as done in a number of papers. What differentiates negative and positive trions are the wavefunctions, not screening, which the authors compute in a way similar to their previous work, Refs.4,49. The problem of Auger processes is quite a bit more complicated beyond lowest order perturbation theory as illustrated for CdSe qdots by Voznyy and co-workers in Phys.Rev.B 84, 155327 (2011) and I would encourage authors to pursue this research.

Response: *The different behavior between positive and negative trions comes from several aspects, including their wave function localization and shapes (as discussed by the referee), as well as the dielectric screening. For clarity, we summarize the main factors affecting Auger recombination rates of trions in Fig. R2, and factors affecting the relative Auger rates of the two types of trions are listed in Table R1. If simple effective mass (or k,p) like method is used, and considering the larger effective mass and higher degeneracy of the hole, usually larger Auger effect of positive trion will be the result. This however is not consistent with the experiment. A more realistic treatment reveals a more complicated picture. Indeed, the hole is more localized, which usually leads to larger Auger effect. On the other hand, although the same dielectric screening function is used for electron and hole, but this is a distance and position dependent dielectric function (not just a constant). Since hole is more localized at the core of the core/shell structure, its screening is larger than the case of electron, which extends to the surface of the QD, where it is not fully screened. We believe this contribute to the fact for some cases, the positive trion Auger rate can be slower than the negative trion, as observed experimentally. As shown in Table R1, screening effect is the sole factor that could make the Auger rate of negative trion larger than the positive one for a CdSe/CdS core/shell QD system. The Auger rate ratio between the positive and negative trions also sensitively depends on the geometry of the QD. With thicker shell, the electron becomes more delocalized, which leads to smaller Auger rate.*

Figure R2. Diagram of factors affecting Auger rate of a trion.

Table R1. The effects of main factors on the relative Auger rates of negative and positive trions for CdSe/CdS core/shell QDs.

	X ⁻	X ⁺	Relative Auger rates (as all other factors being equal)
Density of the final high energy state (DOS)	Low	High	X ⁻ < X ⁺
Effective distance between the two particles $d_{rr'}$	Long	Short	X ⁻ < X ⁺
Screening effect	Weak	Strong	X ⁻ > X ⁺

As for the way we calculate the Auger rate, we actually did not only use the lowest order. Very similar to the PRB 84, 155327 (2011) paper, we indeed have done a CI calculation for the trions using the almost degenerate close-by single electron energy levels. This is then followed by the Auger coupling calculations. So, we are not using a single Slater determinant (single configuration) to describe our trions. This is especially important for the positive trion due to the almost degenerate single particle states. Indeed, for more complicated cases (like biexciton), this becomes an even more important issue. In the current study, we are also using realistic charge patching method and wave functions to describe the electronic states, which is important. Our theoretical results are in quantitative agreement with the experimental results, thus adding weights to both the experimental interpretation for the identity of trions, also the adequacy of our calculation. We now have added some sentences (as below) to clarify the above points (Page 15, Line 350-354), and have also added the above PRB paper in our references (Ref. 59 in the revised Main Text).

“When the initial single-particle states are degenerate or nearly degenerate, a configuration-interaction (CI) expansion of the many-particle states is used to account for the coupling between the nearly degenerate Slater determinants (PRB 84, 155327, 2011). This is especially important for the positive trion due to the almost degenerate single-particle states.”

Comments: Perhaps I missed it but if the authors could unambiguously identify charge of a trion, e.g., with a gate or some other way? and show engineering of Auger rates with different surrounding screening environment, I would be happy to reconsider this paper for publication.

Response: *For identification of charge of a trion, please see responses above. In addition to the verification evidences included in the previous submission, unambiguous assignment of negative trion—the decisive one for the scheme—is included in this revision. The related results are now included as Figure 2 and discussed in the main text. In short, the new results are found to be consistent with the assignment based on the indirect (and also literature) results.*

We agree that, potentially as a general mechanism, it would be interesting to explore dielectric screening with significantly different environment. All results included in the main text can be regarded as the dielectric screening of ligands/solvent environment to the electron/hole wavefunctions with different spatial distribution. Because ligands/solvent environment is organic in nature, it would be complimentary to explore dielectric screening effects of environment with inorganic nature.

To do so, we choose epitaxially grow additional ZnS shells onto the CdSe/CdS core/shell QDs to form CdSe/CdS/ZnS core/shell/shell QDs. Because of the very large band offsets for both valence and conduction bands between CdSe and ZnS, the ZnS shells can be viewed as dielectric environment for both electron and hole. In comparison, electron wavefunction of an exciton would greatly delocalize into the CdS shells, which makes ligands/solvent as an effective dielectric environment for the electron wavefunction of an exciton within a CdSe/CdS core/shell QD. The preliminary results of this new system are summarized as an Appendix attached to the end of this response letter. As expect, strong screening of the ZnS shells reduces Auger recombination rate of the biexciton by half, which increases the biexciton photoluminescence quantum yield by ~200%.

Responses to the comments from Reviewer #2.

Comments: The work is scientifically sound good. The problem is worth of investigation, due to the experimental and especially theoretical motivations. In this paper an experimental investigation

have been demonstrated and agreed with the theoretical studies, but there are a lot of interesting results and comments on them in theory. Related literature is reviewed and obtained results have been compared with it.

Response: *Thanks for encouragement. No action needed here.*

Comments: Despite the above, the authors should clarify some key aspects for the publication of the manuscript, which in my opinion are mandatory.

The following observations should be analyzed carefully and keep in mind the respective changes:

1. I recommend to include other references for theoretical studies of CdSe/CdS core shell QDs.

Response: *We have included a brief introduction on theoretical studies on CdSe/CdS QDs with necessary references (Page 14 Paragraph 3). For your information, it is as follows.*

“The electronic structures of CdSe/CdS core/shell nanocrystals have been studied intensively. It is well established that the valence band maximum (VBM) state is strongly confined in the CdSe core as a result of the large valence band offset between CdS and CdSe (Li Appl. Phys. Lett. 84, 3648, 2004 and Eshet Nano Lett. 13, 5880, 2013). However, their conduction band offset is small (Luo ACS Nano 4, 91, 2010), thus the conduction band minimum (CBM) confinement in the core/shell structure is much weaker (Schrier Phys. Rev. B, 73, 245332, 2006). Depending on the shape and size of the core and shell, the band alignment can be either type-I or quasi-type-II (Li Appl. Phys. Lett. 84, 3648, 2004, Eshet Nano Lett. 13, 5880, 2013 and Schrier Phys. Rev. B 73, 245332, 2006). Theoretical studies also revealed that the Auger processes for positive and negative trions in CdSe/CdS QDs are asymmetric. From tight-binding calculations, Sargent et al. reported that in a CdSe/CdS QD with 4 nm core and 10 nm shell, Auger recombination can be six times faster for positive trions compared to negative ones (Jain Nano Lett. 16, 6491, 2016). Using the $k \cdot p$ method, Efros et al. showed that increasing the CdS shell thickness can lead to a much stronger suppression on the Auger recombination for negative trion than for positive trion (Vaxenburg Nano Lett. 16, 2503, 2016). These calculations agree with some of our experimental results.”

Comments: 2. In the manuscript the authors analyze the experimental properties of CdSe/CdS core shell QDs that deals with the theoretical studies. What is missing is a description of results in the literature on different geometries depend dielectric screening that were already studies and a comparison with the observed results.

Response: *In the revised version of the manuscript, we have added the following descriptions in the text (Page 15 Paragraph 3):*

“Actually, the dielectric constant can change with the quantum dot size (Wang Phy. Rev. Lett. 73, 1039, 1994). Theoretically, one approach is to describe the reduction of the overall dielectric

constant as a consequence of the increase band gap in the QD. For example, Efros et. al. investigated the intraband and interband Auger processes in CdSe/CdS core/shell QDs by introducing reduced dielectric screening as an input parameter in the calculation. The best agreement (Vaxenburg Nano Lett. 16, 2503, 2016) between theory and experiment was obtained with the effective dielectric constant being ~40% of the bulk semiconductor. Alternatively, the reduced overall dielectric constant can be described as a reduction of dielectric response for the locations near the inorganic-organic interface (Cartoixa Phys. Rev. Lett. 94, 236804, 2005). Such microscopic picture of the dielectric response allows us to develop a location dependent dielectric function model which is used in the current work.”

Comments: 3. A phonon carrier studies is recommended in this paper, the study of exciton properties were done - what is missing is a description on the phonons interactions, the interface on phonons play an increasingly large role to modify the size in QDs, so I suggest to add a paragraph about the phonon carrier interactions for the studied QDs.

Response: We like to thank the referee for the suggestion. Indeed, the electron-phonon interaction is important for the carrier dynamics in quantum dot. In other studies, we are simulating phonon-assisted carrier cooling etc. In the current studies, the Auger interaction happens after the carriers are completely cooled, thus it is in the ground state of a trion. One possible effects of phonon for the Auger itself, is the phonon-assisted Auger process. In the current study, that phonon effect is represented by a finite broadening of the Fermi golden rule in the Auger calculation. We do find that, such broadening is important in order to get smooth and reliable results. In the revised version of the manuscript, we have added the following paragraph in the text (Page 19 Paragraph 4 and Page 20 Paragraph 2):

“In terms of non-radiative carrier dynamics, two elementary processes are important. One is the carrier cooling assisted by electron-phonon coupling after excitation, and the other is the electron-electron interaction induced Auger recombination. In the current experiment, the measured photoluminescence decay dynamics is in the nanosecond time range after the initial excitation. This is far beyond the initial hot carrier cooling, which typically happens within a few picoseconds. We can thus assume the Auger processes start from the ground state of a trion.

There is a phonon-assisted Auger process (Hyeon-Deuk J. Phys. Chem. Lett. 5, 99, 2014 and Wen J. Appl. Phys. 118, 015702, 2015). Within this process, not only the electron (or hole) can be excited to their higher energy states through the electron-electron interaction, but phonons can be absorbed or emitted. However, these additional channels do not change the overall oscillator strength of the Auger process. Instead, they just spread out the original zero phonon Auger line, thus broadening the Auger peak. In the current study, this effect has been approximated by using a finite broadening of the energy delta function in the Fermi golden rule in Eq. S6. We have used a peak width of 10 meV (close to kT) to represent this effect. Further increasing the peak width does not significantly change the results.”

Responses to the comments from Reviewer #3.

Comments: This manuscript reports measurements of Auger recombination rates in core/shell quantum dots with different core and shell sizes. Excellent results are obtained through a combination of synthesis of samples with outstanding quality and careful, thorough measurements. Biexciton, negative-trion, and positive-trion Auger rates are separately determined, and the novel observation is made that the positive- and negative-trion rates depend differently on shell thickness, with particular geometries enabling a reversal of the usual ratio of Auger rates. The explanation in terms of dielectric screening is convincing, has to the best of my knowledge not been properly considered before, and provides a new tool to enable control over Auger rates. I recommend publication of the manuscript in Nature Communications.

Response: Thanks for encouragement. No action needed here.

Comments: The only significant comment that I have on the manuscript is that quality of written English is poor in several places. The manuscript should be carefully proofread by a native English speaker or a professional editing service before it can be published.

Response: Thanks for reminding us. We tried our best to improve the English throughout the paper.

REVIEWERS' COMMENTS:

Reviewer #1 (Remarks to the Author):

The Authors addressed all my comments, in particular they added experiments identifying the sign of the trion, i.e., negative trion. I feel that the manuscript is much improved and I recommend publication.

Reviewer #2 (Remarks to the Author):

I am writing to confirm that comments were modified and now I feel that the paper is suitable for publication in Nature Communications. I do emphasize that this decision has to do with the quality of the work and the modifications that were made. The authors added the missed references and they improved the theoretical and experimental motivations and the phonon studies.